# EFFICACY OF LANGUAGE MODEL SELF-PLAY IN NON-ZERO-SUM GAMES

## ABSTRACT

Game-playing agents like AlphaGo have achieved superhuman performance through self-play, which is theoretically guaranteed to yield optimal policies in competitive games. However, most language tasks are partially or fully cooperative, so it is an open question whether techniques like self-play can effectively be used to improve language models. We empirically investigate this question in a negotiation game setting known as Deal or No Deal (DoND). Crucially, the objective in DoND can be modified to produce a fully cooperative game, a strictly competitive one, or anything in between. We finetune language models in self-play over multiple rounds of filtered behavior cloning in DoND for each of these objectives and evaluate them in self-play and in collaboration with humans. We find that language models improve substantially in self-play, achieving 14-17× higher scores in task reward after finetuning. Further, the trained models generalize to both cooperation and competition with humans, scoring 2.5-6× higher than base models. We view these results as an early promising sign for language model self-play in cooperative settings, despite a lack of theoretical guarantees.

## 1 INTRODUCTION

Many of the greatest achievements in artificial intelligence have occurred in two-player zero-sum (2p0s) games such as Go (Silver et al., 2016), chess (Silver et al., 2018), and heads-up poker (Brown & Sandholm, 2018). One key technique enabling these breakthroughs has been *self-play*, in which identical copies of a model are pitted against each other and used to generate new training data. By iteratively training on their own data from games of self-play, models like AlphaGo were able to continue improving long past the threshold of human performance. In certain types of 2p0s games, self-play is theoretically guaranteed to produce optimal policies, given sufficient model capacity and compute (Bai & Jin, 2020; Bai et al., 2020). However, in settings that involve collaboration with humans, self-play is no longer guaranteed to yield optimal policies (Strouse et al., 2021).

It is an open question whether the same principles that led to the success of models like AlphaGo can be applied to language models. Empirically, previous work on training agents to communicate via self-play has shown that they invent uninterpretable communication strategies (Kottur et al., 2017); even when initialized with natural language data, self-play can cause models to gradually diverge from human-interpretable language (Lewis et al., 2017). As a result, much work has focused on mitigating these challenges, e.g., by regularizing with models trained on human data (FAIR, 2022).

In this work, we examine the effect of *game objectives* on self-play between language models. We run a series of experiments on a negotiation task known as Deal or No Deal (Lewis et al., 2017) and train language models for multiple rounds of self-play across three different objectives on this task, ranging from fully cooperative, to semi-competitive, to strictly competitive. Contrary to expectations, we find that self-play leads to large improvements in both the cooperative and semi-competitive settings. These results generalize to human experiments, where scores improve by up to 2.5× in the cooperative setting and 6× in the semi-competitive setting. In contrast, we find minimal improvements in the strictly competitive setting, where models tend to overfit during self-play.

We then investigate the reasons behind these improvements, finding that models trained with self-play better follow task instructions, hallucinate less, and obtain a higher agreement rate with humans. However, at the same time, self-play causes model dialogues to become less diverse and does not appear to teach high-level strategic reasoning or negotiation tactics in our experiments. Although

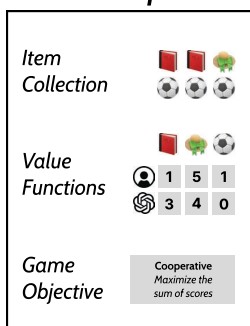
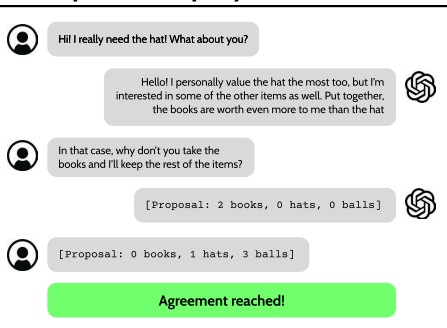
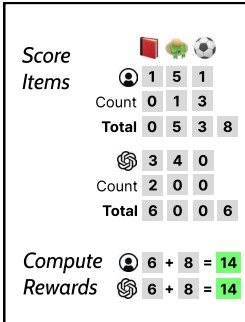

Figure 1: We ran experiments on a modified version of the Deal or No Deal negotiation game from Lewis et al. (2017). In this game, two players are presented with a shared collection of items and private value functions over those items. Players can send messages to each other and then each submit private proposals describing the items they wish to receive. If the proposals are compatible, then the items are scored. In our modified version of the task, players may receive reward based not only on their own item scores, but on the item scores of the other player as well. This modification allows us to convert Deal or No Deal into a cooperative or strictly competitive game.

these results highlight potential room for improvement, we view them as a promising initial signal for self-play training of large language models and plan to release all code for our environments, models, and human data collection to support future research in this area: `anonymous.tbd`

## 2 COOPERATIVE AND COMPETITIVE GAMES

*Can language model self-play be effective under both cooperative and competitive objectives?* To address this question, we conducted experiments on Deal or No Deal (DoND; Lewis et al., 2017), a two-player negotiation game in which players decide how to divide a shared pool of items through natural language dialogue. Although introduced as a semi-competitive game, DoND has the special property that it can be readily adapted into either a cooperative or strictly competitive (i.e., zero-sum) game, with minimal modifications to its rules. Below, we describe the rules of DoND, how we modify its objective, and how we convert it into an environment for evaluating language models.

**Game Setup**   Following Lewis et al. (2017), we present two players with a shared collection of books, hats, and balls (with 5-7 total objects). Each player is assigned their own private value function, mapping each item type to an integer point value. Value functions are selected according to the following criteria: (1) each item is valued by at least one player, (2) the maximum score either player can receive is 10, and (3) at most one player can achieve the maximum score. Players must divide the objects; if they fail to reach an agreement, they both receive zero points. These rules ensure that the game is *semi-competitive*: players have conflicting objectives, but if they fail to cooperate at all then they will end up without any points.

**Game Rules**   The game is divided into two phases. In the first phase, players send messages discussing which items they would like to receive. At any point, either player may end this phase by submitting a private proposal, delineating which items they would like to claim from the shared collection. During the second phase, no additional messages can be sent, and the other player must respond by submitting a proposal of their own, which ends the game. If players submit complementary proposals (i.e., adding up to the total number of objects in the shared collection), then players receive rewards according to their respective objectives. Hence, players should aim both to reach an agreement and to optimize the value of that agreement.

**Game Objectives**   In the original formulation, players receive a reward equal to the inner product of their value function and proposed set of objects. However, we observe that this objective can be modified to convert DoND into a cooperative game, a strictly competitive one, or anything in

---

**Algorithm 1** Language Model Self-Play

---

1: **Input:** Language model $M$, number of games per iteration $K$, number of iterations $N$, function exec which runs a game of self-play and returns dialogues and rewards.
2: **Output:** Finetuned language model $M$
3: **for** $n = 1$ to $N$ **do**
4:     Initialize an empty set of dialogues $\mathcal{D}$
5:     Initialize a list of rewards $\mathcal{R} = []$
6:     **for** $k = 1$ to $K$ **do**
7:         ▷ Obtain dialogues and rewards:
8:         $(D_1, D_2, R_1, R_2) \leftarrow \text{exec}(M)$
9:         Add $(D_1, R_1)$ and $(D_2, R_2)$ to $\mathcal{D}$
10:         Append $R_1$ and $R_2$ to $\mathcal{R}$
11:     **end for**
12:     Compute the average reward $\bar{R} = \frac{1}{2K} \sum_{r \in \mathcal{R}} r$
13:     Initialize an empty set $\mathcal{D}_{\text{filtered}}$
14:     **for** each $(D, R) \in \mathcal{D}$ **do**
15:         **if** $R > \bar{R}$ **then**
16:             Add $D$ to $\mathcal{D}_{\text{filtered}}$
17:         **end if**
18:     **end for**
19:     Finetune $M$ using dialogues from $\mathcal{D}_{\text{filtered}}$
20:     **If** early stopping criteria met **then** break
21: **end for**

---

between. For example: if players receive rewards $R_1 := X$ and $R_2 := Y$ in the original setting, instead setting the objective to $R_1 = R_2 = X + Y$ for both players results in a fully cooperative game. More generally, we can define the objective for Player 1 as $R_1 := X + \lambda \cdot Y$ for $\lambda \in [-1, 1]$, and vice versa for Player 2. In this work, we experiment with $\lambda = 0$ (*semi-competitive*), $\lambda = 1$ (*cooperative*), and $\lambda = -1$ (*strictly competitive*), although we note that in principle $\lambda$ can be tuned to smoothly interpolate between these objectives. The maximum reward that can be obtained in a single game is 10 in the strictly and semi-competitive settings and 19 in the cooperative setting.[1]

**Game Environment**    Akin to recent work on language agents (Abdulhai et al., 2023; Lin et al., 2024), we implement an OpenAI Gym-like (Brockman et al., 2016) environment for evaluating language models on DoND. This environment provides affordances for (1) generating new random game instances, (2) prompting language models with game rules and context, (3) handling messages and formal proposal actions, (4) computing player rewards, and (5) sending comprehensive error messages to models in case they violate the game rules, e.g., by sending incorrectly formatted proposals. We provide full details in Appendix A and in our open-source code release.

## 3 LANGUAGE MODEL SELF-PLAY

We begin by evaluating pretrained language models, prompted only with task instructions and the current game context, as detailed in Appendix A. In contrast to prior work, we do not prompt our models with few-shot example dialogues (Gandhi et al., 2023) or finetune them on task-specific data (Lewis et al., 2017) in the majority of our experiments. Doing so helps us avoid biasing models toward specific patterns of behavior. We then finetune these models over many rounds of self-play.

We implement a straightforward algorithm for language model self-play (Algorithm 1) based on filtered behavior cloning (filtered BC; Chen et al., 2020; 2021; Zelikman et al., 2022). In this setting, two language models with identical parameters but different prompts play $K$ games and receive rewards according to their (identical) objectives. Each game produces two dialogue histories, one

---

[1]Reward is obtained by adding the item scores from each player in the cooperative setting. However, the maximum possible reward is 19 and not 20 because of the game's constraint that at most one player can achieve the maximum item score. The maximum *average* self-play reward is 7.5 in the semi-competitive setting and 15 in the cooperative setting; in the zero-sum setting, average self-play scores are always zero. Finally, the average score across all Pareto-optimal outcomes in the semi-competitive setting is 6.6.

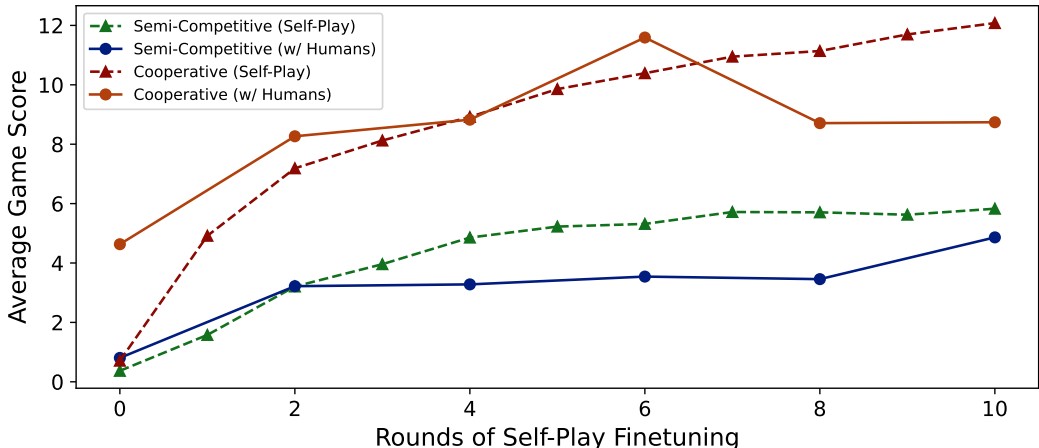

Figure 2: Language model self-play significantly increased model performance in both cooperative and semi-competitive games. Moreover, these results generalized to collaboration and competition with humans, leading to improvements of up to $2.5\times$ and $6\times$ the baseline scores, respectively. We found that human-LM baseline scores were higher in the cooperative setting as humans can help "guide" models to avoid common failure modes.

from each player's perspective. The average score across games is computed, and dialogues with above-average scores are kept and used for finetuning the model. This procedure is repeated for $N$ iterations or until early stopping. We set $K = 500$ and $N = 10$ for the majority of our experiments.

We ran experiments on GPT-3.5 (`gpt-3.5-turbo-0125`; Ouyang et al., 2022), using the OpenAI API for finetuning. At the time of experimentation, GPT-3.5 was the most capable model for which finetuning was publicly available. We also ran preliminary experiments with open-weight models such as Mixtral 8x7B and LLaMA-2-70B, but we found that they did not achieve enough nonzero scores to improve from the first round of self-play. Finally, we ran a small-scale baseline experiment on GPT-4 (`gpt-4-turbo-2024-04-09`; OpenAI, 2023). Due to the high cost of experiments, we leave investigation of other models to future work.

## 4 HUMAN EXPERIMENTS

To evaluate whether model improvements generalize beyond self-play, we built a web interface which allows humans to play DoND against our trained models. We evaluated models on both the cooperative and semi-competitive objectives across the timecourse of training. However, due to the large number of models we trained and the cost of human experiments, we only ran human evaluation on *every other* iteration of self-play finetuning.

**Crowdsourcing** We ran human evaluation on Amazon Mechanical Turk. After a prescreening survey and three pilot studies, we identified a group of 60 reputable English-speaking workers and invited them to participate in our task. In order to incentivize high-quality dialogue, we primarily compensated workers with bonus pay: each worker earned $1.00 for picking up the HIT, $0.10 for each game played, and $0.20 for each point earned. In total, we collected 1,175 human-LM dialogues, with the average worker receiving pay of $37.50. More details on crowdsourcing, including screenshots of our web interface, can be found in Appendix C.

## 5 RESULTS

### 5.1 BASELINE MODELS

We first evaluated language models without any self-play training. Because we did not provide few-shot example dialogues, GPT-3.5 performed relatively poorly, obtaining a mean score of 0.4 in the

original, semi-competitive setting and 0.7 in the cooperative setting. In contrast, GPT-4 achieved much higher mean scores of 4.3 in the semi-competitive setting and 8.8 in the cooperative setting. Although we use GPT-4 as a reference point for model performance, we did not conduct further experiments on it due to the lack of public finetuning access at the time of experimentation.

GPT-3.5's low scores can primarily be attributed to its inability to consistently reach complementary proposals with itself in self-play, reaching a valid agreement in only 6.8% of games across both objectives. Additionally, this baseline model relies heavily on error-handling from the environment to send messages properly and fails to remain grounded in the game's context throughout an entire dialogue, often hallucinating new items, changes in its value function, or both.

In collaboration with humans, GPT-3.5 obtained a much higher average score of 4.6. While errors tend to compound in self-play, we observed that humans can help "guide" models to avoid common failure modes in the cooperative setting, resulting in higher scores (e.g., by suggesting which objects to propose). However, similar improvements did not occur in the semi-competitive setting, where humans are less incentivized to help models perform well; in the semi-competitive setting, the baseline GPT-3.5 model achieved a mean score of 0.8.

## 5.2 SELF-PLAY FINETUNED MODELS

Despite weak performance of the initial models, language model self-play was highly effective, as shown in Figure 2. When evaluated against another copy of the same model, self-play finetuning increased scores by as much as $14\times$ in the semi-competitive setting ($0.4 \rightarrow 5.8$) and $17\times$ in the cooperative setting ($0.7 \rightarrow 12.1$). Although these experiments were conducted on GPT-3.5, these scores are significantly higher than those of a baseline GPT-4 model, as reported in Section 5.1.

Improvements from self-play generalized to collaboration and competition with humans as well, with scores increasing by $6\times$ in the semi-competitive setting ($0.8 \rightarrow 4.9$) and $2.5\times$ in the cooperative setting ($4.6 \rightarrow 11.6$). In the cooperative setting, we note that human-LM scores peaked at 11.6, but began to decline before the 10th iteration of self-play. We do not report scores for any model after the 10th iteration, as they tended to stabilize or even decline.

**The Case of Strict Competition** Between fully rational agents, communication is not useful in a 2p0s game (Crawford & Sobel, 1982). Additionally, for the strictly competitive setting, due to the zero-sum nature of the game, it is not informative to report mean scores in self-play, as the average score in this setting will always be zero. We instead evaluated the quality of trained models based on how well they performed against a separate model, GPT-4. Additionally, due to a sparsity of positive-scoring games, we modified the filtering criteria for strictly competitive self-play to also include samples from zero-scoring games in which a valid agreement was reached. While models improved at reaching valid agreements in self-play, we found they generalized poorly against other agents. Our preliminary results indicated that even the best-performing models for this objective would routinely fail to reach agreements with humans, so we instead ran 100 games between each iteration's model and GPT-4, confirming that the model failed to improve outside of self-play. We report results for this objective, along with further implications, extensively in Appendix D.

## 5.3 COMPARING THE EFFECT OF SELF-PLAY TO TASK-SPECIFIC FINETUNING DATA

We also considered the case where our initial model was finetuned on an externally provided corpus of task-specific data. For the semi-competitive objective, we finetuned models on 300 nonzero scoring games from the original human-human dataset in Lewis et al. (2017). However, because task-specific data only exists for the original task formulation, we used GPT-4 to generate a comparable amount of finetuning data for the cooperative objective; to enable fair comparison, we also used GPT-4 to generate finetuning data for the semi-competitive objective. We finetuned GPT-3.5 on the nonzero scoring games for each of these three settings and then repeated the self-play algorithm from Algorithm 1, providing a finetuned GPT-3.5 model as input instead of the baseline GPT-3.5.

We found that finetuning drastically improves the performance of base models, as shown in Table 1. Applying self-play training on top of finetuned models provided slight additional gains, ranging from +6% ($11.0 \rightarrow 11.7$) in the cooperative setting to +38% ($4.2 \rightarrow 5.8$) in the semi-competitive setting. These improvements are much smaller than the improvements from self-play finetuning on

| Model | GPT-4 (Cooperative) | | GPT-4 (Semi-Comp.) | | Human (Semi-Comp.) | |
|---|---|---|---|---|---|---|
| | Self-Play | Human Eval | Self-Play | Human Eval | Self-Play | Human Eval |
| GPT-3.5 | 0.7 | 4.7 | 0.4 | 0.7 | 0.4 | 0.7 |
| Finetuned | 11.3 | 11.0 | 5.5 | 4.2 | 5.3 | 4.8 |
| Iteration 1 | 11.0 | **11.7** | 5.5 | 4.7 | 6.0 | 5.3 |
| Iteration 2 | **11.4** | **11.7** | **5.8** | **5.8** | **6.2** | **5.7** |
| Iteration 3 | 10.8 | 10.5 | 5.6 | 3.6 | 6.1 | 5.5 |

Table 1: Mean scores of models initially finetuned on task-specific data, which was either generated using GPT-4 self-play or extracted from prior human experiments in Lewis et al. (2017). After training, models were evaluated both in self-play and via human experiments. While model scores improved in the earliest iterations of self-play finetuning, performance plateaued and even declined much earlier than in the experiments without initial training on task-specific data.

| | Agreement | | | | Pareto-Optimality | | | |
|---|---|---|---|---|---|---|---|---|
| | Semi-Competitive | | Cooperative | | Semi-Competitive | | Cooperative | |
| | Self-Play | Human | Self-Play | Human | Self-Play | Human | Self-Play | Human |
| Before | 6.8 | 13.3 | 6.8 | 32.7 | 2.2 | 8.9 | 5.8 | 16.3 |
| After | **96.4** | **76.5** | **91.0** | **64.0** | **46.0** | **49.0** | **89.6** | **38.0** |

Table 2: Agreement and Pareto-optimality rates (%) before and after ten rounds of self-play finetuning. Models show significant improvements across objectives in both self-play and human generalization. We observe a relatively lower agreement rate for human performance in the cooperative setting, as well as the remaining headroom in Pareto-optimality, especially in human collaboration.

base models. However, we note that: (a) our finetuning process may be viewed as a single round of filtered BC where zero-scoring games are filtered out, (b) self-play provides further gains on top of task-specific finetuning, and (c) task-specific finetuning data may not be available for all tasks. Therefore, we view finetuning not as a replacement for self-play, but as a complementary method.

# 6 ANALYSIS

## 6.1 ERRORS, AGREEMENTS, AND PARETO OPTIMALITY

In order to understand the effectiveness of language model self-play, we analyzed the frequency of errors and successful agreements over the course of training. We also analyzed the rate of Pareto-optimal outcomes, i.e., those where neither player's score can improve without reducing the other's.

Prior to self-play finetuning, GPT-3.5 frequently made errors such as submitting invalid proposals or sending messages and proposals at the same time. When this happened, the game environment would provide an error message, as detailed in Appendix A, and the model would be given another chance to generate a message. With the baseline model, errors were generated in 14% of semi-competitive games and 56% of cooperative games. If the model received five error messages in a row, the game aborted, and both players received a score of zero; this outcome occurred in just 1% of semi-competitive games but in 22% of cooperative games. After self-play finetuning, errors occurred in less than 1% of games, suggesting that models effectively learned the game rules. We present additional results on error and abort frequencies in Appendix E.

Models also improved in their ability to achieve valid agreements and Pareto-optimal game scores, both in self-play and in human experiments, as shown in Table 2. Trained models achieved almost perfect agreement rates in self-play, with 96.4% of games ending in an agreement in the semi-competitive setting. While the agreement rates are lower in collaboration with humans, this can partially (but not wholly) be attributed to human error. For example, we observed that humans sometimes failed to read overly long messages in full, resulting in failed proposals; however, such errors may also be viewed as the fault of models failing to communicate efficiently.

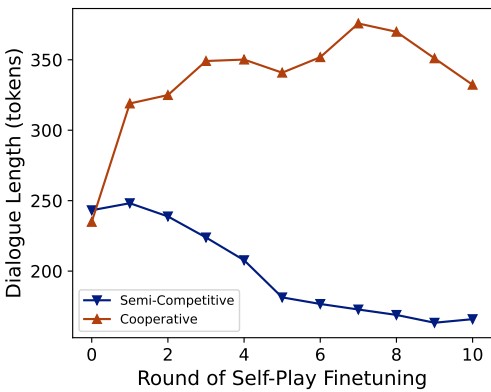 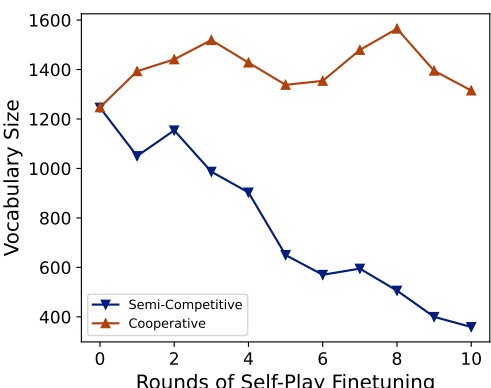

Figure 3: Mean dialogue lengths (left) and aggregate vocabulary sizes (right) for every model iteration, for both semi-competitive and cooperative objectives. Dialogues under the semi-competitive objective progressively shrank in length, while dialogues under the cooperative objective grew significantly longer. Similarly, in the semi-competitive setting, vocabulary size trended downward, but the model maintained and even expanded its vocabulary when trained with the cooperative objective.

We found that model improvements after self-play can primarily be attributed to an increased rate of agreement. To justify this claim, we calculated Pearson correlation coefficients between completed iterations of self-play and scores achieved, before and after filtering out samples that failed to reach a valid agreement. For our self-play data under the semi-competitive objective, this yielded $\rho = 0.44$ before filtering and $\rho = 0.34$ after. On human-LM data with the same objective, however, the correlation drops from $\rho = 0.29$ to $-0.04$ after filtering out non-scoring games. In other words, although self-play scores appear to rise after filtering out games which fail to reach a proposal, these improvements do not generalize to humans.

We hypothesize that the increased agreement rate can primarily be attributed to better understanding of task instructions, following the environment rules, and not hallucinating items or proposals, rather than the acquisition of strategic negotiation or optimization behavior. As further evidence for this claim, we note that even after self-play finetuning, models routinely missed opportunities for better scores: only 49% of semi-competitive games with humans resulted in Pareto-optimal outcomes, and only 38% of cooperative games did so. As a result, we note that there is still substantial headroom on this task, although we expect techniques other than filtered BC may necessary to close this gap.

## 6.2 HALLUCINATIONS AND GROUNDING

We observed that baseline models often failed to reach agreements because they hallucinated items, lied about their point values, or made proposals which contradicted their previous utterances; in contrast, self-play finetuned models rarely seemed to hallucinate. To quantify this claim, we used a stronger language model to annotate the rate at which messages or proposals in self-play dialogues exhibited inconsistencies. Specifically, we prompted GPT-4 to indicate whether each message (1) lied about the player's point value for an item, (2) made an impossible proposal based on the item counts in the game context, or (3) made a proposal explicitly contradicting what was agreed upon

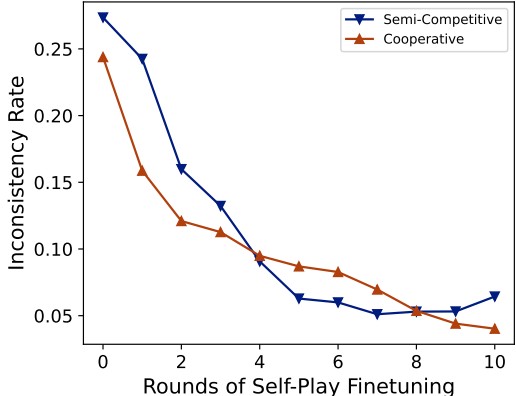

Figure 4: The rate of hallucinations or otherwise inconsistent messages and proposals declines over the course of self-play finetuning. We report this value as a per-message rate, rather than per-game.

in the discussion.[2] We found that the rate of
inconsistent messages decreased from 27% to
6% in semi-competitive games and from 24%
to 4% in cooperative games. In contrast to prior work, which found that self-play increased the rate
of lying and deceptive behavior (Lewis et al., 2017), these results suggest that producing mostly
honest and accurate messages is a reasonable strategy, even in not-fully-cooperative scenarios. We
provide further details, including the prompt used to generate these results, in Appendix E.

## 6.3 DIALOGUE LENGTH AND DIVERSITY

One natural question is whether self-play finetuning has any adverse effects on language quality.
We qualitatively observed that dialogues in the semi-competitive setting became less diverse over
the course of self-play finetuning. We quantified this in two ways: (1) by average dialogue length
and (2) by the number of unique words produced during each iteration, as reported in Figure 3.

We first computed the average dialogue length over 500 hundred games of self-play during each
iteration of model training in the semi-competitive and cooperative settings. We found that self-play
caused dialogues to become substantially *longer* in the cooperative setting but *shorter* in the semi-
competitive one. We hypothesize that this discrepancy may occur because agents in the cooperative
setting are incentivized to share all information; qualitatively, we often observed models sharing
exact details of their private value functions. Additionally, we found that models which argue with
each other for too long are more likely to "go off the rails" and fail to reach an agreement at all;
because these games receive scores of zero, this behavior may be filtered out over the course of
self-play under the semi-competitive objective.

We also computed the vocabulary size of each iteration by counting the number of unique word types
produced during 500 games of self-play. We observed a similar trend of decreasing vocabulary size
over the course of self-play in the semi-competitive setting, supporting the hypothesis that the semi-
competitive objective leads to convergence in model behavior. However, as shown in Section 5.2,
these models performed well in both self-play and in human generalization experiments, suggesting
that they may not *need* very diverse communication strategies to achieve high scores. We include
additional results on $n$-gram entropy in Appendix E, where we observe mostly similar trends.

## 7 RELATED WORK

**Grounded Dialogue** Much prior work on goal-oriented dialogue has focused on collaborative
settings. In tasks such as Cards (Djalali et al., 2011; Potts, 2012), CerealBar (Suhr et al., 2019),
OneCommon (Udagawa & Aizawa, 2019), and DialOp (Lin et al., 2024), two or more agents must
collaborate via natural language dialogue to achieve a shared goal within an environment. In many of
these tasks, models are often evaluated via self-play, which serves as a proxy for human evaluation.

Another line of work has focused on the case where agents have conflicting goals. A handful of
grounded dialogue tasks are focused on bartering or negotiation, including Deal or No Deal (DoND;
Lewis et al., 2017), which is based on the multi issue bargaining task from DeVault et al. (2015),
as well as CaSiNo (Chawla et al., 2021) and the fruit trading game from Gemp et al. (2024). These
games are all structurally similar and differ primarily in the number and types of objects they use,
as well as the public availability of human data. In the Craigslist Bargaining task (He et al., 2018),
agents negotiate on the price of a object for sale. Recently, several new game environments have
been proposed for benchmarking language model agents (Chalamalasetti et al., 2023; Qiao et al.,
2023; Li et al., 2023; Wu et al., 2024a; Gong et al., 2024). Fried et al. (2023) provides additional
discussion of grounded dialogue tasks and modeling approaches.

**Self-Improving Language Models** Lewis et al. (2017) trained GRU-based language models on
the Deal or No Deal task using REINFORCE (Williams, 1992). In contrast to our work, Lewis et al.
(2017) did not learn a model *tabula rasa* but instead interleaved reinforcement learning from self-
play with supervised learning on task-specific data to avoid divergence from human-interpretable
language. Divergence issues abound in other settings where models are trained via self-play, such

---

[2]In order to validate this method, the authors manually annotated 100 randomly selected messages across
iterations and compared their predictions with those of GPT-4, obtaining 92% agreement.

as in emergent communication (Kottur et al., 2017; Lowe et al., 2019; Tomlin & Pavlick, 2019), including some work on negotiation settings (Cao et al., 2018; Noukhovitch et al., 2021).

A wave of recent work has focused on methods for autonomously improving large language models at training (Ouyang et al., 2022; Bai et al., 2022; Abdulhai et al., 2023) or inference (Shinn et al., 2023; Yao et al., 2023; Wu et al., 2024b) time. Methods like StaR (Zelikman et al., 2022) and Rest-EM (Singh et al., 2024) iteratively train models on their own filtered outputs to improve reasoning capabilities. Closely related to our work is Pan et al. (2024), which iteratively trains models for device-control tasks using filtered behavior cloning; however, in contrast to our work, Pan et al. (2024) studies a single agent interacting with an environment, rather than multiple agents interacting with one another. Another closely related paper is Fu et al. (2023), which uses self-play to refine language models for a distributive bargaining task; in contrast to our work, Fu et al. (2023) use in-context learning rather than iterative finetuning, leading to less major performance improvements. Finally, the recently proposed SOTOPIA-$\pi$ (Wang et al., 2024) trains models via a similar filtered BC method on a benchmark of social tasks (Zhou et al., 2024).

**Multi-Agent Reinforcement Learning** Training agents against copies of themselves is a long-standing technique in reinforcement learning (Littman, 1994), popularized in the past decade by models like AlphaGo (Silver et al., 2016) and AlphaZero (Silver et al., 2017). Experiments on games such as Overcooked (Carroll et al., 2019) and Hanabi (Bard et al., 2020) have shown that policies learned via self-play often fail to generalize to collaborative or imperfect information games. Methods such as fictitious self-play and population play have been proposed to address these issues (Heinrich et al., 2015; Strouse et al., 2021), but have primarily been applied in games without language components. In games with language, a KL-regularization objective is often used to prevent language from drifting too far from human-written training data (Jaques et al., 2019; FAIR, 2022).

## 8 CONCLUSION & DISCUSSION

Our experiments showed that language model self-play can lead to significant performance improvements in both semi-competitive and cooperative games. This finding contradicts existing wisdom that self-play is ineffective in collaborative domains (Strouse et al., 2021), or that models need to be trained on task-specific human data to avoid divergence from human-interpretable language (Lewis et al., 2017; FAIR, 2022). One hypothesis is that because we observed significant model improvements after just ten rounds of self-play, the model may not have had time to overfit to cooperation with itself. Another hypothesis is that better language models might be more robust to the negative effects of self-play. Given a model with good generalization abilities, finetuning on self-play games might be able to elicit model capabilities which are not directly present in the self-play data. Furthermore, self-play with pretrained language models might actually function more similarly to population play, since large language models are trained on text from a population of users and may simulate different personas in different contexts (Pataranutaporn et al., 2021; Park et al., 2023).

Although game scores increased significantly after self-play, this increase can be almost entirely attributed to an increase in the percentage of completed games, rather than better strategic reasoning or negotiation tactics. We anticipate that future work may be able to obtain even larger improvements by combining self-play with approaches other than filtered BC, such as natural language reflections (Shinn et al., 2023). Another possible approach is described by Srivastava et al. (2024), in which a language model is used to describe distributional differences between good and bad trajectories.

Finally, the effectiveness of methods like self-play is completely dependent on a reward signal, which in this work was obtained from the game environment. To apply similar methods in real-world settings, we anticipate that models will need to rely on feedback from general-purpose, learned evaluators (e.g., as in Du et al., 2023; Pan et al., 2024). We leave further investigation of the challenges associated with bringing self-play into real-world application domains to future work.

## REPRODUCIBILITY STATEMENT

We describe our method in Section 3 and Algorithm 1, and we provide a complete description of our environment implementation in Appendix A. We provide the prompts we used in Figure 5 in Appendix A and Figure 9 in Appendix E, and describe model hyperparameters in Appendix B. We

provide full details of our crowdsourcing experiment in Appendix C. Code for all results in this paper, including model training, human experiments, and analysis, will be provided upon publication. Additionally, we will release our environment code for DoND in order to facilitate future work on this task and language model self-play in general.

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

## A   ENVIRONMENT IMPLEMENTATION

We implemented an environment for the task under which language models can play the game with each other or against another human. For initializing a new instance of the game, we sample from a list of 4,186 valid game contexts (shared item counts and private value functions for each player), provided by Lewis et al. (2017). The environment then takes turns prompting each player to either send a message (prepended by [message]) or submit a proposal (prepended by [propose]). After detecting a submitted proposal, the environment forces the other player to submit a proposal of their own. This is enforced by error correction, as described below. Once a game is completed, the environment uses the game context and the submitted proposals to determine the final score of each player, conditioned on the objective under which the players were instructed to play.

**Error Correction**   This environment comes with comprehensive error-handling to correct models' errant outputs. Specifically, the environment will reply with instructions for correcting errors when errors are detected, providing the model with the opportunity to format its output correctly. The errors that we check for, and their corresponding correction messages, can be found in Table 3. If a model generates five errant outputs in a row, then the environment aborts the game, and both players receive zero score.

| Error | Correction Prompt |
|---|---|
| Outputting text without a prefix of either "[message]" or "[propose]" | `Your output should either begin with [message] or a [propose].` |
| Submitting proposals before any messages have been sent | `Please begin the dialogue by discussing how you'll divide the items before submitting a private proposal.` |
| Sending messages with multiple mentions of "[message]" or "[propose]" (i.e. outputting multiple messages in a row) | `Do not include any mentions of [message] or [propose] after the initial prefix. Please just send a single message, beginning with [message].` |
| Sending messages after a proposal has been submitted | `Opponent's proposal must be followed by a proposal of your own.  Please send a proposal, beginning with [propose].` |
| Submitting proposals with incorrectly sequenced items | `Item counts must be sequenced in the following order:  books, hats, and then balls.` |
| Submitting proposals with more than three item counts | `There should only be counts for three items in your proposal:  books, hats, and balls.` |
| Submitting proposals with invalid item counts, based on game context | `Item counts suggested are invalid based on game context; some of your proposal's item counts are greater than total items available.` |

Table 3: Our game environment sends error messages to language models if they produce ill-formed outputs, e.g., sending a message after the discussion phase ends. The model then has an opportunity to send a new message based on the correction. If the model repeatedly fails to produce well-formed outputs, then the game aborts and both players receive zero score.

**Zero-Shot Prompting**   Across every setting, the initial models are zero-shot prompted with the game's rules and the instructions for sending messages and submitting proposals with the correct syntax. The choice of this approach over few-shot prompting was motivated by the concern that few-shot examples might influence strategies chosen by the model during inference. Our preliminary experiments found that models would closely match the negotiation techniques used in few-shot examples; for example, if prompted with dialogues where players shared their exact value functions, the model would consistently share its own values, whether doing so was advantageous or not.

**Prompting with Conversation History**    Following the format recommended by the OpenAI API's chat completions endpoint, the prompt containing game instructions is sent under the `system` role; for subsequent dialogue, any messages sent by the model itself are categorized with the `assistant` role, and messages from the other player are appended to a model's input as messages from the `user` role. The system prompt used for the semi-competitive objective can be found in Figure 5; other prompts are available in our code release.

## B    MODEL TRAINING AND HYPERPARAMETERS

All models were finetuned using the OpenAI API, with parameters `n_epochs=3`, `batch_size=1`, and `learning_rate_multiplier=8`. For model inference, we generated outputs with `temperature=1`. These parameters were all default values chosen by the OpenAI API, except for the learning rate multiplier, which defaults to 2. Our preliminary experiments with default learning rate multipliers yielded models that not only failed to improve but also devolved significantly in quality. We hypothesize that this occurred because parameters are set dynamically based on the quantity of finetuning data. Because language model self-play requires sequential rounds of finetuning, it may be important to choose initial parameters based on the expectation of future finetuning rounds.

To ensure that GPT-3.5's low initial scores were not a result of suboptimal prompting, we manually crafted 10 alternative semi-competitive prompts (and their cooperative counterparts) and ran 100 games of self-play using GPT-3.5 for each one. For the semi-competitive setting, the mean score across prompts was 0.382, with a minimum of 0.18 and a maximum of 0.52. For the cooperative setting, the mean score across prompts was 0.636, with a minimum of 0.25 and a maximum of 1.17.

## C    DETAILS OF HUMAN EXPERIMENTS

### C.1    GAME INTERFACE

We developed a web interface for human data collection, shown in Figure 12, Figure 13, and Figure 14. The interface provides comprehensive instructions describing the different game modes, as well as an explanation of the bonus pay structure and a running count of bonus pay earned so far. At the end of each game, players see a popup with the number of points and bonus pay earned. If players receive no points (e.g., due to game error or non-compatible proposals), they receive a small amount of bonus pay and an explanation of what went wrong. During the main phase of data collection, players were allowed to complete up to 40 games; after each game, players were given the option to end the HIT and collect bonus play or keep playing.

### C.2    CROWDSOURCING

We ran human evaluation through Amazon Mechanical Turk (MTurk). We restricted our task to workers from the United States with a 98+% HIT approval rate and at least 500 completed HITs, based on recommendations in Huang et al. (2023). In order to filter out bots and low-quality workers, we ran a brief prescreening survey which asked workers to (1) to answer a question about text on a linked, external website and (2) write a 2-3 sentence description of their favorite MTurk task. The authors then manually reviewed responses to the prescreening survey and chose approximately 15% to invite to the main task.

We ran three pilot studies before launching our main human evaluation, with 10 workers each. After the initial pilots, we modified the interface and incentive structure to obtain higher-quality dialogues. We reviewed data from each pilot and removed low-quality workers and spammers from later rounds of data collection. In total, we invited 60 workers to our final round of data collection, although a small number of workers declined the HIT. We include data from the third pilot in our results because we did not modify the game after that point. Figure 6 provides additional statistics on the total number of workers hired.

```
You are an expert in negotiation.  You are about to play a game
with another player.  In this game, you and your partner will
divide a shared set of books, hats, and balls.  Each item has a
point value for you, but you don't know your partner's values.
At the start of the game, you will be given the total number of
objects of each type, as well as your own private value function,
which tells you how many points each object is worth to you.  Your
points will be equal to the sum of item values for all items you
receive.  Your objective is to maximize your points.

On each turn, you can either send a message to the other player,
or submit a private proposal for how to divide the items.  Your
partner will do the same, and both proposals will remain hidden
from each other.  Please push back on any suggestions made by your
partner that you believe would leave you with an unsatisfactory
point total.  However, if the number of items in the combined
proposals don't match the total number of items, both players score
0.

Messages should be formatted like this:
[message] Your message here.

Proposals should be formatted like this:
[propose] (x books, y hats, z balls)

The numbers x, y, and z should be your own item counts.  The item
counts must be whole numbers; you cannot split singular items.  For
example, if you want 1 book, 2 hats, and 0 balls, you would send:
[propose] (1 books, 2 hats, 0 balls)

When discussing, do not leave any of the items unclaimed.  You and
your partner must submit proposals that collectively add up to the
total item counts.  To achieve a nonzero score, your partner would
need to write a complementary proposal that adds up to the total
number of items.  For example, if the total number of items is 3
books, 2 hats, and 1 ball, your partner would need to send:
[propose] (2 books, 0 hats, 1 balls)

Any message that you send should begin with either "[message]"
or "[propose]".  All proposals are final, so make sure that both
players agree about which items are being taken by which player
before ending the discussion with a proposal.

Each message should end with "[END]".

Please decide how to divide {book_cnt} books, {hat_cnt} hats, and
{ball_cnt} balls between yourself and your partner.  This should be
an open discussion; you should only propose after exchanging a few
messages.
To you, books are each worth {book_val}, hats are worth {hat_val},
and balls are worth {ball_val}.
You don't know your partner's item values.
Remember, your goal is to maximize your own score while also
ensuring that your partner will agree to the deal.
```

Figure 5: System prompt used for the *semi-competitive* objective. Values in {brackets} are filled in based on the game context (i.e., item counts and private value functions).

## C.3 Incentive Structure

We paid $1.00 for picking up the HIT and $0.10 per game completed. The majority of pay was distributed through bonuses. We paid a bonus of $0.20 per point earned in the semi-competitive setting and $0.10 per point earned in the cooperative setting, since scores in the cooperative game are on average twice as high. We also paid workers $0.25 in cases where models aborted due to repeatedly generating ill-formed messages.

In contrast, Lewis et al. (2017) paid workers $0.10 per game and $0.05 in bonus pay *only when workers achieved the maximum score of ten points*. We found that this approach incentivized workers to end the game as quickly as possible, as maximizing the number of games played was more lucrative than attempting to achieve a high score.

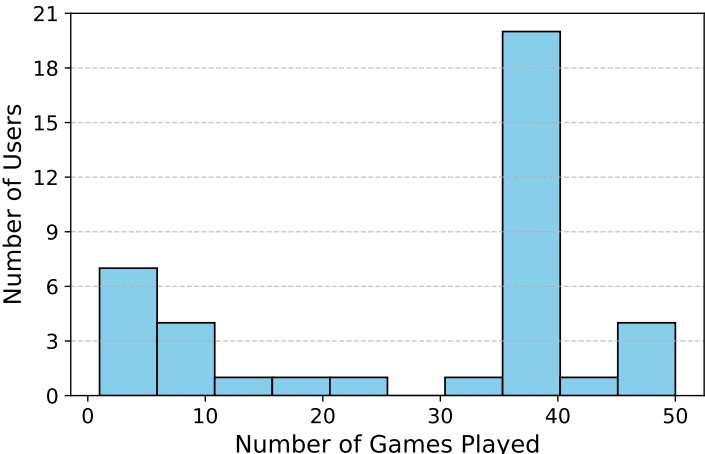

Figure 6: The majority of our workers played the maximum number of games, with a small handful contributing data from both the pilot and the main study. The mean pay for workers was $35.70.

## D Results for Strict Competition

When we applied LM self-play to Deal or No Deal under the strictly competitive objective, the model failed to improve its performance, instead learning strategies that adversely impacted its ability to perform outside of self-play. Since the mean score in self-play for every iteration will always be zero (because the game is zero-sum), we instead evaluated the quality of model self-play through agreement rates. As shown in Figure 7 and Figure 8, while agreement rate trends show that the model's performance in self-play improves, these models fail to generalize to competition with other models, such as GPT-4. Our preliminary human experiments also showed that the model failed to reach agreements in roughly 95% of games.

In our qualitative analysis of successive iterations of the model under the strictly competitive objective, we found that it learned to replicate an inverted proposal strategy; specifically, the model learned a strategy where it submits a proposal based on what the *opposing* player should receive, rather than what the model itself should receive. While the model optimized this strategy in self-play well enough to arrive at valid agreements at a competitive rate with itself, this strategy does not generalize to competition with humans or GPT-4. We found this to be a consequence of the aggressive nature of the strictly competitive objective leading to a smaller proportion of games ending in valid agreements to derive reward signal from. With a smaller number of samples providing reward signal, we risk an outcome where the few samples that are isolated for finetuning achieve their non-zero reward through undesirable strategies (inverted proposals, in this instance) that do not generalize well to human interaction.

Our experiments under this objective illustrate an important takeaway regarding the failure modes of LM self-play. For this strategy to be effective, there must be confidence that the initial model is

capable of achieving high performance through desired strategies with significant probability. Additionally, we speculate that this strategy is most effective in environments with continuous reward.

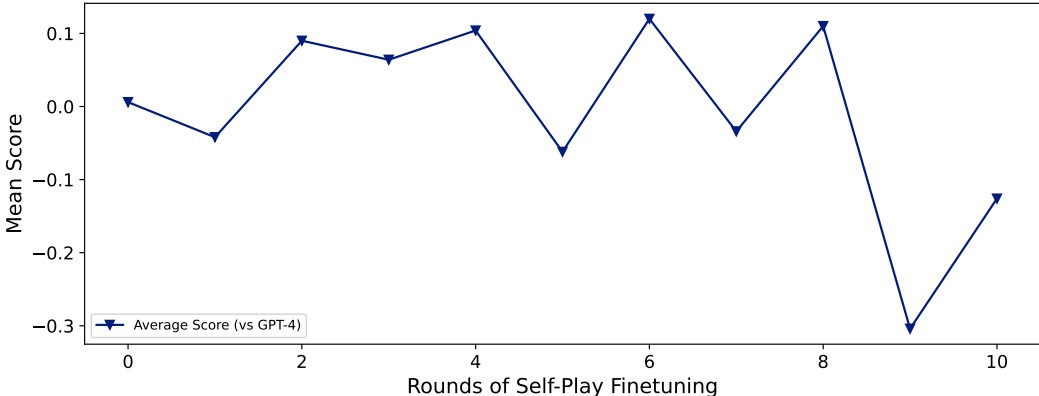

Figure 7: We evaluated the strictly competitive model against GPT-4, since self-play scores are not informative for zero-sum games. However, we determined that the model experienced no significant improvement in generalization.

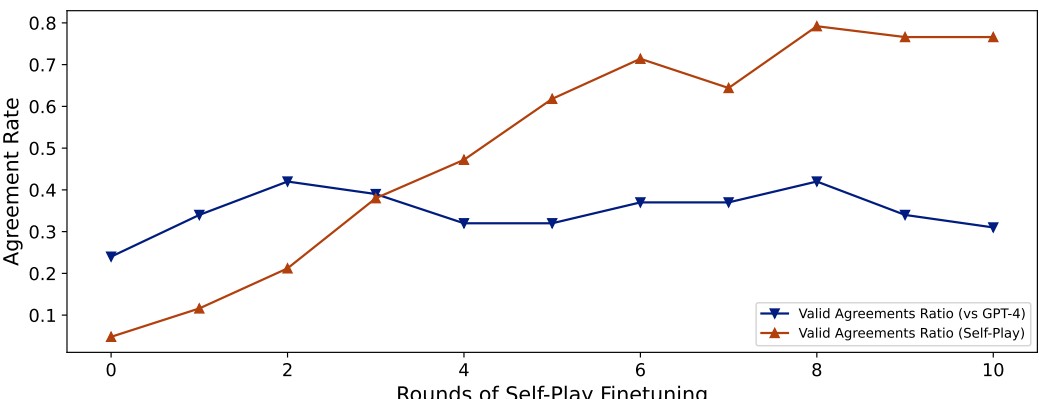

Figure 8: Under the strictly competitive objective, self-play finetuning increased the frequency of games reaching valid agreements in self-play, but the strategies learned generalized poorly to interaction with other agents, such as GPT-4. Our preliminary experiments also indicated a low agreement rate with human competitors.

# E   ADDITIONAL ANALYSIS DETAILS

**Additional Results**   We include additional error and abort rates in Figure 10. To further quantify the diversity of self-play finetuned models, we also fit $n$-gram models on the dialogues produced during each iteration of self-play and report their results in Figure 11.

**Details of Hallucination Analysis**   We used GPT-4 to analyze the frequency of hallucinations and inconsistent messages or proposals in self-play dialogues. Our analysis prompt, provided in Figure 9, takes in an entire dialogue with one player's private value function masked out and returns a list of binary classifications describing whether each message or proposal was consistent with the game context or not. We ran this prompt on 500 games from each iteration of self-play, for each of the semi-competitive and cooperative objectives. For each dialogue, we ran the analysis prompt twice, once from each player's perspective. The results of this analysis are presented in Figure 4.

```
This is a game where players with private value functions negotiate
how to divide a shared set of items and end the dialogue by making
private proposals about which items they will claim for themselves.
The proposals are private and final, meaning the model should
submit proposals that are consistent with what the players agreed
upon.  Analyze the following dialogue between LLMs and determine if
the messages of the dialogue from Player {player_to_analyze} either:
- lies about the player's point value for an item,
- makes an impossible proposal based on the item counts in the game
context, or
- makes a proposal explicitly contradicting what was agreed upon in
the discussion

Your output should be a structured JSON output, containing three
things for each message:
- "message":  a repeat of the line (including the source, e.g.
"Player {player_to_analyze}")
- "analysis":  a step-by-step analysis of whether the message
violates any of the conditions above
- "decision":  a final [YES] or [NO] answer.

Dialogue:
{conversation_history}
```

Figure 9: Prompt used for computing the rate of hallucinations and inconsistent messages or proposals in self-play dialogues. We used GPT-4 to classify whether each message or proposal was consistent with the game context so far, finding that consistency increased over the course of self-play finetuning. Values in {brackets} are filled in based on the game context.

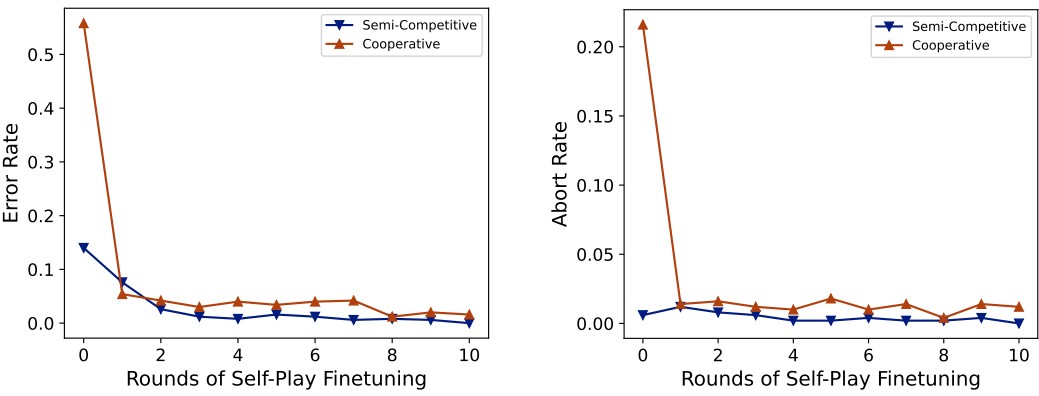

Figure 10: Rate of self-play games consisting of errors (left) or aborts (right) over the course of self-play finetuning. When a model produces an invalid message or proposal, it receives an error message from the environment and is given an opportunity to re-generate the message. If a model makes five errors in a row, the game aborts. Errors and aborts decline over the course of training.

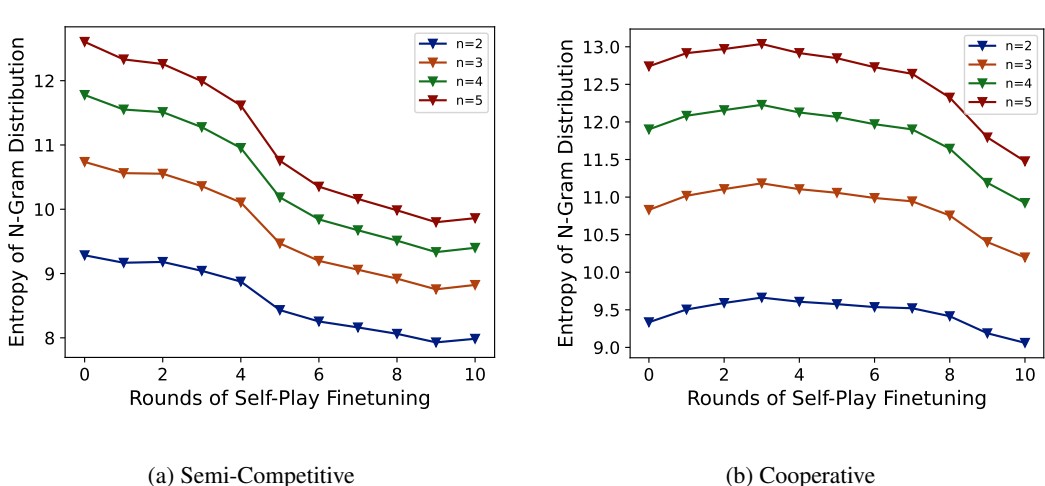

(a) Semi-Competitive          (b) Cooperative

Figure 11: We fit $n$-gram models on the semi-competitive (left) and cooperative (right) self-play games and computed the entropy of their distributions. As with vocabulary size, the entropy decreased in the semi-competitive setting but stayed roughly constant in the cooperative setting.

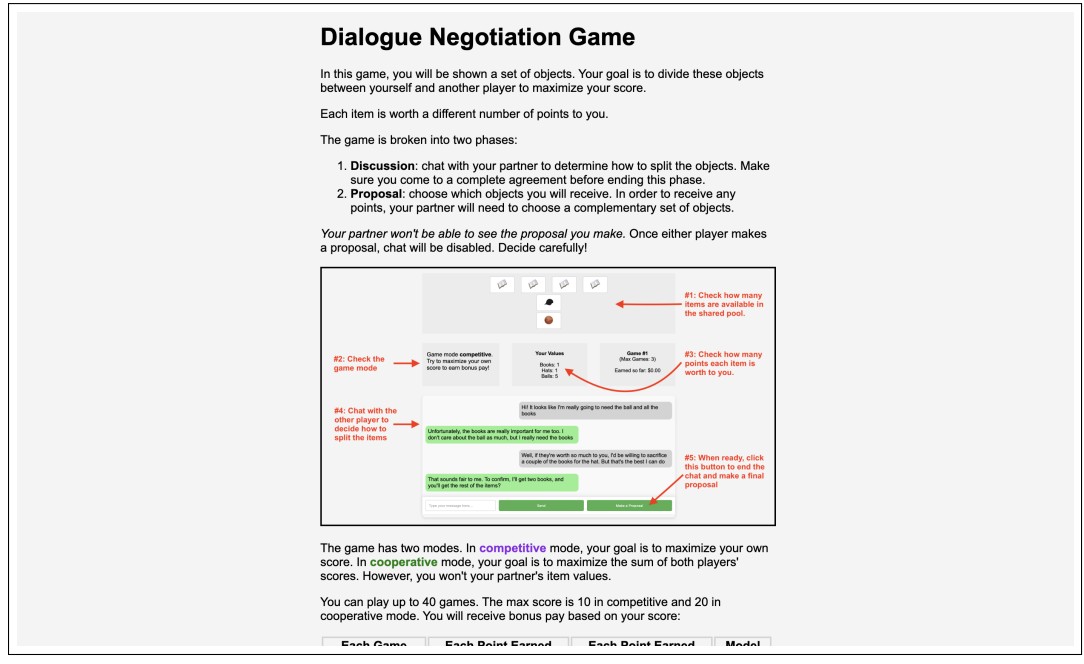

Figure 12: The landing page for our human data collection site provides comprehensive game rules and instructions and an explanation of the bonus pay structure.

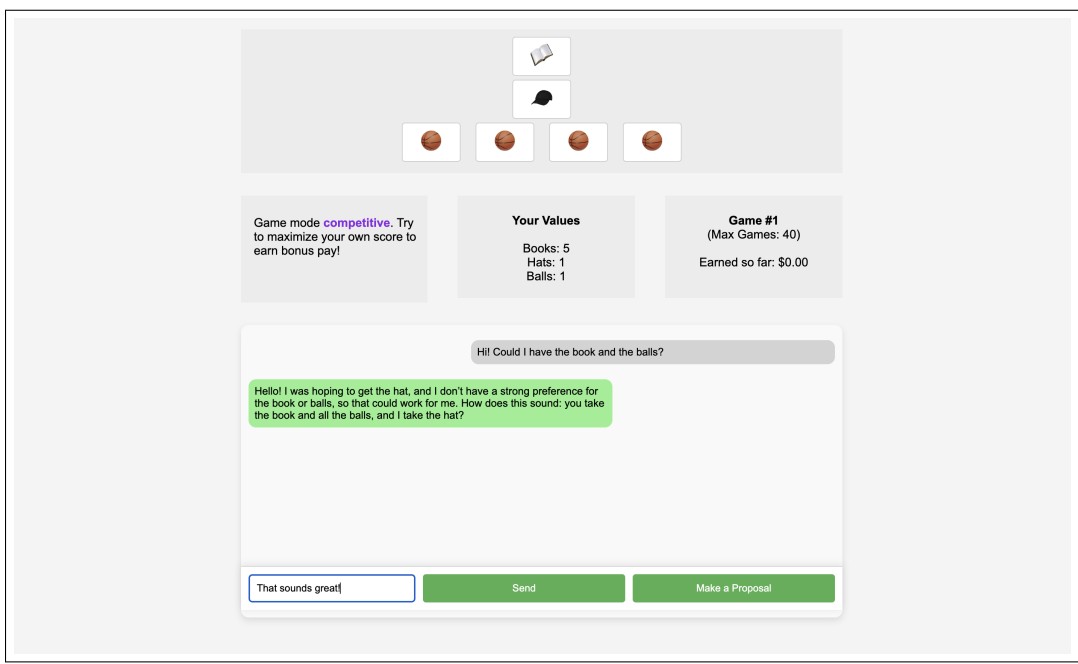

Figure 13: The game interface for human data collection shows the shared item pool, game mode, item values, and a chat window, as well as the number of games played so far and a running count of bonus pay earned.

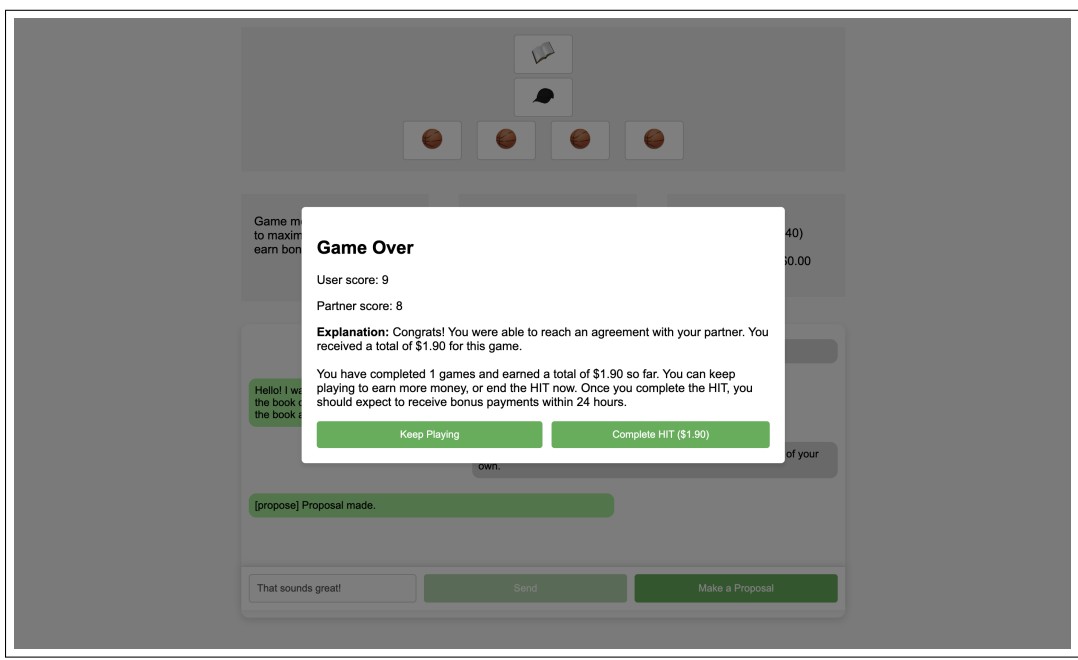

Figure 14: At the end of each game, players see a popup with the number of points and bonus pay earned. Players have the option to end the game and collect their bonus pay or keep playing, up to the maximum of 40 games.

