# OpenReview forum: "Efficacy of Language Model Self-Play in Non-Zero-Sum Games"
_ICLR.cc/2025/Conference — Submitted to ICLR 2025_

### Official Review · Reviewer_qWFo · 2024-11-01

**Soundness:** 3
**Presentation:** 3
**Contribution:** 2
**Rating:** 3
**Confidence:** 4

**Summary:**

This work studies whether self-play can improve language models' (LM) performance in cooperative and semi-cooperative games. They use Deal or No Deal (DoND) as the testbed and find self-play based on filtered behavior cloning significantly improves LMs' performance and generalizes to gameplay with humans.

**Strengths:**

1. Good clarity: the manuscript is well-organized and easy to follow. The DoND game and the idea of language model self-play are clearly described.
2. Good analysis and discussion: the authors give a detailed analysis of the effect of self-play in the DoND task. They also discuss the limitations and directions for future improvement.

**Weaknesses:**

This work studies the question of "whether techniques like self-play can effectively be used to improve language models (in cooperative tasks)". My main concern is that the current manuscript mainly presents preliminary empirical results and lacks solid and sufficient evidence to support a conclusion. Below is the detailed discussion.

1. Limited insights on why and how to make language model self-play work. Self-play has been known to be effective in non-language-based cooperative settings like MPE [1], SMAC [2], GRF [3], etc., so it is expected to work in language-based cooperative settings. Some previous work, as mentioned in the paper, finds that the self-play of language model can diverge from human-interpretable language, while this work finds it leads to performance improvement in DoND. Instead of just presenting these empirical findings, a more important question is to understand why language model self-play works in some tasks but fails in others, and how to make it work in more general tasks. However, this work mainly provides a limited and hypothetical analysis of this question in DoND only. It would be more insightful if the authors could investigate more general settings to provide an in-depth understanding behind the empirical findings.
2. Potential overclaim due to limited task, model, and training. This work investigates only one task (DoND) with one LM (GPT-3.5) and trained for just ten rounds. This is more like a case study and it would require evidence on more tasks with more LMs and training for more rounds to support the authors' claim that "self-play leads to large improvements in both the cooperative and semi-competitive settings" (L047-048). In fact, the authors also mentioned some preliminary results that may negatively impact the current claim.
    1. More models: as mentioned in L190, Mixtral 8x7B and LLaMA-2-70B fail to improve from the first round of self-play. This shows that self-play can lead to no improvement in some LMs. Providing results of Mixtral 8x7B and LLaMA-2-70B can examine the soundness of the claim.
    2. More training: as mentioned in L258-259, the authors "do not report score for any model after the 10th iteration as they to stabilize or even decline". If the performance can decrease with more training rounds, is it possible that, with more self-play rounds, the model will still diverge to uninterpretable language? It would be better if the authors could provide results of more iterations to examine the long-term stability of language model self-play.
    3. More tasks: as mentioned in L466-467, the performance increase in DoND "can be almost entirely attributed to an increase in the percentage of completed games, rather than better strategic reasoning or negotiation tactics". In fact, cooperative DoND may be too easy because the agents can simply share their values for each item and reach the optimal decision. The current results mainly show that self-play can improve LM's understanding of a specific task, instead of reasoning or decision-making ability. It would be more convincing if the authors could provide results in other more complex reasoning and decision-making tasks.

Therefore, I think the current manuscript mainly presents preliminary empirical results and lacks solid and sufficient evidence to support its claim.

[1] Lowe, Ryan, et al. "Multi-agent actor-critic for mixed cooperative-competitive environments." Advances in neural information processing systems 30 (2017).

[2] Samvelyan, Mikayel, et al. "The starcraft multi-agent challenge." arXiv preprint arXiv:1902.04043 (2019).

[3] Kurach, Karol, et al. "Google research football: A novel reinforcement learning environment." Proceedings of the AAAI conference on artificial intelligence. Vol. 34. No. 04. 2020.

**Questions:**

Please see Weaknesses.

---

> ### Author Response · Authors · 2024-11-28
>
> Thanks for your review! We would like to make a couple of clarifications:
>
> > Self-play has been known to be effective in non-language-based cooperative settings like MPE [1], SMAC [2], GRF [3], etc., so it is expected to work in language-based cooperative settings
>
> We respectfully disagree with this claim. In particular, past work (including the work cited here) has shown that self-play in cooperative settings can lead to models which collaborate well with themselves, but not to models which collaborate well *with humans*. See [4] for further discussion on this point
>
> > Instead of just presenting these empirical findings, a more important question is to understand why language model self-play works in some tasks but fails in others, and how to make it work in more general tasks.
>
> For sure, this would be a great contribution. Our best guess is that self-play works in this setting because LLMs can be finetuned with relatively few samples, avoiding the overfitting described in [4]; however, you are correct that it would require experiments on additional domains to justify this hypothesis. Because language model self-play is a computationally expensive method, however, this would be a relatively difficult set of experiments to run. We view our work as more of a “proof of existence” that vanilla self-play can be used to train LLMs which can better collaborate with humans, which has not been shown before.
>
> > This is more like a case study and it would require evidence on more tasks with more LMs and training for more rounds to support the authors' claim that "self-play leads to large improvements in both the cooperative and semi-competitive settings" (L047-048).
>
> We don’t believe this is an overclaim – we aren’t claiming that self-play leads to large improvements in all cooperative and semi-competitive settings, only in the ones we studied.
>
> > More models, more training, more tasks
>
> We totally agree that all of these directions are valuable. Unfortunately, as discussed in the response to R2, the total cost for all computational experiments in this paper (excluding human data collection) is about $6000, making it intractable to run large-scale experiments on additional models and tasks.
>
> > If the performance can decrease with more training rounds, is it possible that, with more self-play rounds, the model will still diverge to uninterpretable language?
>
> This is totally possible. As with many training methods, it’s possible to overfit, and the assumption is that you should conduct early stopping before this divergence occurs.
>
> [4] Collaborating with Humans without Human Data. https://arxiv.org/abs/2110.08176

---

> > ### Comment · Reviewer_qWFo · 2024-12-02
> >
> > Thank you for your response. I think we have reached an agreement that the current manuscript serves as an early sign for language model self-play using GPT-3.5 in the game of DoND. However, as I discussed in my review, the current result is limited to a single LM in a single task with limited rounds. It is possible that the conclusion no longer holds for a different model, or on a different task, or using more training rounds. I understand that more experiments are more expensive, but these results are necessary evidence to support a solid conclusion about the efficacy of language model self-play in non-zero-sum games. Therefore, I'm not convinced by the current results and keep my score as is.

---

### Official Review · Reviewer_J6ic · 2024-11-04

**Soundness:** 3
**Presentation:** 4
**Contribution:** 3
**Rating:** 5
**Confidence:** 4

**Summary:**

This paper explores the application of self-play training to language models in non-zero-sum settings through experiments in the Deal or No Deal negotiation game. The authors modify DoND to create cooperative, semi-competitive, and strictly competitive game objectives. Empirical results show substantial improvements in model performance across cooperative and semi-competitive settings in self-play and interactions with humans, but limited gains in strictly competitive scenarios. The study also identifies increased instruction adherence and reduced hallucinations as benefits of self-play while noting limitations in strategic behavior learning.

**Strengths:**

1. The paper provides extensive quantitative results demonstrating significant performance improvements. And results show models trained with self-play perform better in collaboration with humans.
2. Includes detailed analysis of errors, agreement rates, and Pareto optimality.
3. Provides detailed insights, such analyses of dialogue length and hallucination rates.
4. The paper is clearly presented and easy to follow.

**Weaknesses:**

1. In cooperative games involving humans, performance declines as the number of rounds increases from 8 to 10. Why does this occur? This trend is not seen in other game settings.
2. The experiments and analysis are limited to a single game, which could introduce bias. Evaluations across additional environments are needed for validation.
3. The study only considers cooperative and semi-competitive settings. Including a wider range of competitive and cooperative levels could provide deeper insights into the impact of cooperation on the effectiveness of self-play.

**Questions:**

1. What could be the underlying reasons for the observed performance decline in cooperative games involving humans after 8 to 10 rounds? Why is this trend not present in other game settings?
2. What alternative environments/games could be explored to strengthen the study's findings?
3. Would expanding the study to include more competitive and cooperative levels provide a more comprehensive understanding of the impact of cooperation on self-play efficacy?

---

> ### Author Response · Authors · 2024-11-28
>
> Thanks for the positive review! In response to your concerns:
>
> > What could be the underlying reasons for the observed performance decline in cooperative games involving humans after 8 to 10 rounds? Why is this trend not present in other game settings?
>
> We found that our model began to overfit after the sixth iteration of cooperative self-play. In particular, in certain settings, the model would exhibit an “inverse proposal” strategy where it described one set of objects and then formally proposed the complementary set. As a result, the model’s scores increased in self-play, but the human-LM scores declined. In the semi-competitive setting, overfitting doesn’t occur until a later iteration.
>
> > What alternative environments/games could be explored to strengthen the study's findings?
>
> One challenge here is finding games which can be naturally converted into cooperative and competitive versions. Possible options include the fruit trading game from [1], or CaSiNo in [2], although both are structurally very similar to the Deal or No Deal domain. However, we did not run experiments on these additional domains due to cost.
>
> > Would expanding the study to include more competitive and cooperative levels provide a more comprehensive understanding of the impact of cooperation on self-play efficacy?
>
> This is definitely possible! We did not explore this direction due to cost and resource constraints, and because the cooperative and semi-competitive results were relatively similar. However, as noted in Section 2, these experiments can straightforwardly be run by changing the value of \lambda.
>
> [1] States as Strings as Strategies: Steering Language Models with Game-Theoretic Solvers. https://arxiv.org/abs/2402.01704v2
> [2] CaSiNo: A Corpus of Campsite Negotiation Dialogues for Automatic Negotiation Systems. https://arxiv.org/abs/2103.15721

---

### Official Review · Reviewer_9sE8 · 2024-11-04

**Soundness:** 4
**Presentation:** 3
**Contribution:** 4
**Rating:** 6
**Confidence:** 4

**Summary:**

This paper explores finetuning LLMs with self-play data in the negotiation game "Deal or No Deal" (DoND). The authors implement a filtered behavior cloning on LLMs and achieve promising results. They also build a web interface which allows humans to play DoND against their trained models. The experiments demonstrates that their method enables LLMs to achieve substantial performance gains across various game objectives—ranging from cooperative to competitive settings. The study also highlights improvements in task adherence and reduced hallucination rates, suggesting the models became more aligned with game rules over time.

**Strengths:**

- A key innovation of this work is the self-play training on large language models, which I believe is a novel approach. While self-play has been successfully used in other domains with smaller models, applying it to LLMs in a negotiation game is interesting. This novel approach expands the scope of self-play, making it as a promising technique for training LLMs in complex, dialogue-based tasks.
- The experimental results demonstrate substantial improvements, with the performance of the models increasing multiple times through finetuning.
- The modified DoND game environment in this paper is highly engaging for evaluation across different interaction types, including cooperative and competitive objectives.
- The authors plan to make their dataset and code publicly available, which may encourage further research in the field.

**Weaknesses:**

- The study exclusively focuses on language models and does not incorporate any reinforcement learning (RL) baselines for comparison. Including RL-based models could have provided a broader benchmark, highlighting the unique contributions of self-play for LLMs while also revealing potential strengths or weaknesses relative to established RL techniques.
- The self-play data generation and subsequent finetuning require substantial computational resources, resulting in high training costs. The authors mentioned themselves in the paper that they only ran a small scale baseline experiment on GPT-4 due to its high cost.

**Questions:**

- Can you further elaborate why the performance in iteration 3 is worse in Table 1? Do you have any examples showcasing the overfitting?
- What do you think is the reason why open source LLMs cannot improve their performance with your method?
- How long does it take to finetune one round/iteration?

---

> ### Author Response · Authors · 2024-11-28
>
> Thank you for your positive review! To answer your questions:
>
> > Can you further elaborate why the performance in iteration 3 is worse in Table 1? Do you have any examples showcasing the overfitting?
>
> It’s hard to give definitive answers, but we can provide a best guess based on our qualitative evaluation of the dialogues. The main reason seems to be that models trained on GPT-4 outputs are overly agreeable, and after a few rounds of self-play training these models become pushovers. For example, in this dialogue, the model accepted a deal worth zero points:
>
> ```
> User: [message] I need the hat and ball and at least one book
> Model: [message] I value the books at 0, so I'm happy to let you have one or all of them. Since you need the hat and the ball, would it be agreeable if I take the remaining books? How many points are the hat and the ball worth to you? This information can help us make a fair division.
> User: [message] deal, you get two books and I get two books, the hat and the ball
> Model: [message] That works for me. I'll take two books, and you'll take two books, the hat, and the ball. Shall we proceed with finalizing this proposal? Or is there anything else you'd like to adjust or discuss?
> User: [propose] Proposal made. You must now respond with a proposal of your own.
> Model: [propose] (2 books, 0 hats, 0 balls)
> ```
>
> > What do you think is the reason why open source LLMs cannot improve their performance with your method?
>
> In short: language model self-play only works when models are above a certain baseline level of performance. At the time of experimentation, most open source models lagged significantly behind closed source models, especially on agent tasks [1]. During the first round of self-play, we found that open source models obtained scores of 0 in almost every game, resulting in a lack of data for finetuning. This result parallels other research in self-improving LLMs, which has also found that strong base models are necessary to see gains from self-improvement [2].
>
> > How long does it take to finetune one round/iteration?
>
> About one hour, using GPT-3.5 via the OpenAI API.
> ***
> In response to your other concerns:
>
> > The study exclusively focuses on language models and does not incorporate any reinforcement learning (RL) baselines for comparison. Including RL-based models could have provided a broader benchmark, highlighting the unique contributions of self-play for LLMs while also revealing potential strengths or weaknesses relative to established RL techniques.
>
> Could you please elaborate further on what type of RL baselines you have in mind here? Ultimately, the task we consider is a grounded dialogue game, so we do need a model that is capable of generating language. While that language model could be trained with more sophisticated RL techniques (e.g., DPO), we chose to focus on filtered behavior cloning as we wanted the simplest possible experimental setup, in order to avoid possible confounds.
>
> > The self-play data generation and subsequent finetuning require substantial computational resources, resulting in high training costs.
>
> For sure, this is absolutely true. The approximate cost of running experiments for this paper (including exploratory experiments, but not counting human evaluation) was about $6000. This is the main reason we did not experiment with a wider range of domains or models. However, as time goes on, we anticipate that the cost of language model finetuning and inference will decline, making self-play and related approaches more feasible. Additionally, high training costs may sometimes be permissible in industry, e.g., if a model only needs to be trained once but will be deployed to millions of users.
>
> [1] AgentBench: Evaluating LLMs as Agents. https://arxiv.org/abs/2308.03688
> [2] Autonomous Evaluation and Refinement of Digital Agents. https://arxiv.org/abs/2404.06474

---

### Official Review · Reviewer_eGtk · 2024-11-05

**Soundness:** 2
**Presentation:** 2
**Contribution:** 3
**Rating:** 5
**Confidence:** 3

**Summary:**

This work proposes to study how LLMs learn in competitive games. They choose to investigate using mostly GPT-3.5 and the game Deal or No Deal (DoND) where two players negotiate to split a set of items.

They find that GPT-3.5 is unable to play the game using just prompting but through iterations of supervised finetuning on successful self-play data, it learns to improve its play. Improvement are seen not just in self-play but also when evaluated playing with humans. Warm-starting from a GPT-3.5 model finetuned on task-specific data does even better.

Analyzing the improvements, models are largely just learning the rules and making valid proposals, not strategically negotiating. Still, iterative self-play does reduce the rate of hallucinating wrong preferences and inconsistent. As expected, cooperative play increases the length and diversity of messages (as more information can always be used) while semi-competitive play reduces both (perhaps to reduce exploitability).

**Strengths:**

I really enjoy the research direction and perspective of the authors. Agentic LLms are becoming more popular and their interactions in non-cooperative environments are understudied, so this work has a novel and interesting approach.

The authors do well to demonstrate where their self-play is effective, especially the result that it learns to reduce hallucinations. They also provide interesting analysis of human-AI play and different ways in which training affects results. There are a lot of interesting insights to be gained from this paper and I give the authors some slack for attempting something quite novel and difficult (LLM self-play). It is clear that there are many challenges in this direction (LLaMA-70B was not good enough at initialization!) so all reasonably positive results are encouraging.

**Weaknesses:**

The major weaknesses of this paper is the lack of explicit understanding of their own game dynamics, which are crucial for understanding expected behaviour, baselines on performance, and conclusions that can be drawn. This is likely because of missing insights from previous, non-LLM work. The overall story of the paper is also muddled and so it is unclear what are the main insights and whether the experiments support the conclusions. I believe this is fixable in the review period and give my recommendations for what the authors can improve.

Overall, I think the authors should formalize the competitiveness of the game, explain why they study competitiveness since their results seem negative, and give a more full, coherent story. If the authors answer one of the three major weaknesses here I will increase my score and if they answer all three I will increase my score again.

### How competitive is your game?

First and foremost, the authors claim to work on semi-competitive games but never actually measure the competitiveness of their game! This is crucial, as proven theoretically in the famous work by [Crawford and Sobel (1982)](https://www.jstor.org/stable/1913390), the level of achievable communication is inversely proportional to the level of competition in the game. Communication is not useful in a fully competitive game. This was further shown to be true in neural networks learning to communicate (emergent communcation) in semi-competitive games by [Noukhovitch et al (2021)](https://arxiv.org/abs/2101.10276) which also looks at an extremely similar negotiation game [(Cao et al, 2018)](http://arxiv.org/abs/1804.03980)

The competitiveness of DoND depends on the specific items and preferences each player has over the items. A game with $\lambda=1$  can be highly competitive if both players have similar values for the same items, highly cooperative if they value different items, or a mix of the two. Without knowing this, it is impossible to say whether a score of 5 is good or not. At minimum, the authors should calculate the average competitiveness of the game and have a line in Figure 2 for the average pareto-optimal score in the semi-competitive game.

You show "Pareto-Optimality" in Table 2 but this term is never defined or the value formally stated. I believe this to be the score for each player if the items were to be divided such that players had as high and as similar possible scores (i.e. a "fair" pareto optimal). This should be formalized.

As well, the that your game is "cooperative, competitive, or anything in between" in your abstract is not necessarily true. You should show this as done in Noukhovitch et al.

**There should be no communication in a fully competitive game**

As proven by Crawford and Sobel, and shown empirically by Noukhovitch et al, a fully competitive game should have no communication. Indeed the best action for your DoND agents with $\lambda = -1$ is to not communicate and fail to agree to any deal as that will allow them to get score 0.

Your experiments trying to achieve valid agreements makes no sense as it is explicitly advantageous to *not* reach agreements. As argued by Noukhovitch et al, when there is no incentive to communicate, agents will not communicate.

**What is optimal play?**

To figure out if your models are actually learning the rules of the semi-competitive game you should outline what optimal play really is. Noukhovitch et al (2021) argues that the negotiation game is an ultimatum game which would give first-mover advantage and mean the first player should win most of the points. The rounds of dialogue in-between are cheap talk that shouldn't affect this game-theoretic optimal play, but I am open to other interpretations.

### What's the point of semi-competitive and self play?

If the actual gains in your setting come about from learning to play the game and follow the rules (Section 6.1, 6.2) then what is the point of competition in your setting? Why does it matter if the game is competitive if all you're doing is learning to play?

Further, [Ross et al (2024)](https://arxiv.org/abs/2408.02784v1) show that LLMs are generally altruistic. Based on this analysis, they should not be playing the competitive game effectively (at least at initialization) as they would seek to maximize fairness for their opponent. This suggests to me that your competitive scenario is not even maximizing the competitive objective and your results seems to suggest this too.

You should make a strong argument for whether self play and competitive games are a negative result and why it matters. This is especially important since GPT-4 seems to play the game reasonably well at initialization so it is unclear why you're training and not creating a better prompt.

### What is the story and what is actually being learned

The paper feels a lot like a splattering of results without a very coherent backbone. This is fixable and I am also willing to change my view.

Currently the story reads as
- Here's a challenge for LLMs: DoND and a solution: iterative finetuning with LLM self-play data
- Here's how it improves over a very basic prompt baseline
- Here's an analysis that shows that it's really just learning the basic rules of the game
- Here's a slight difference between cooperative and competitive play

This is a bit underwhelming and entirely hinges on the fact that your deal or no deal game is difficult to play using prompts alone. I am not convinced of this. I am especially not convinced that a better prompt could solve the cooperative game with GPT-3.5.

I think you need to reformat this to make it clear how difficult DoND is or, more likely, to argue that perfect prompting of LLMs is difficult and your self-play method is an easier approach to getting complex agentic behaviour. One method (just as an example) would be to actually find the perfect cooperative prompt and show how it fails in the competitive case, demonstrating how much harder it would be to transfer the prompts.

For cooperative play, it is clear you are learning the rules of the game and cooperation. But what is being learned in the semi-competitive game? The competitive game includes the ability to exploit other agents, so are you really learning the game if you're not learning to exploit other agents? This is an interesting negative result that contributes to Ross et al (2024) if you're showing that even training on a competitive game does not easily achieve a competitive but altruistic strategy, but this should be clear.

**Questions:**

**major**

- why do errors occur in 22% of cooperative games but only 1% of semi-competitive? (section 6.1)

- how can you be sure that human-AI performance is not bolstered by humans compensating for your AI playing badly?

**minor**

- why do you talk about fully competitive games in both the intro and game objectives when communication is not useful in 2 player fully competitive games (Crawford and Sobel, 1982)

- Can you change the words "finetuning" in Section 5.3? It is very confusing to differentiate between the "warm-started" model and the act of iterative finetuning towards the task.

- You claim that Humans are to blame for not reading overly long messages but it could be that the blame is with the model for learning to generating long output: this is actually just alignment tax / language drift. As shown in Figure 3 left, is your model drifting away from human-preferred language towards longer useless text when being trained in self-play?

- Why do you include lying about an item preference value in your errors for 6.2? Lewis et al argued that this lying could be advantageous / effective deception.

**references**

- Your method of iterative finetuning with interactive LLMs with filtering is quite close to [Rest-EM](https://arxiv.org/abs/2312.06585), so this probably deserves a citation.

- The negotiation game of Lewis et al (2017) itself based on a negotiation game of [Devault et al (2017)](https://aaai.org/papers/10335-10335-toward-natural-turn-taking-in-a-virtual-human-negotiation-agent/)

- The citation and explanation of how SPIN (Chen et al, 2024) is *not related* feels superfluous and strange. Just don't have it in related work.

- KL is always used to prevent language from drifting too far, CICERO is just a simple case of this. You can cite [Jaques et al (2019)](http://arxiv.org/abs/1907.00456) for this if you'd like. Also, if I'm not mistaken, the language component of CICERO was trained a bit separately from the game-playing component which is the real way it didn't drift too far from initialization.

- Why are you citing Chen et al (2021) for GPT-3.5 in Section 3? I think this is a mistake

---

> ### Author Response · Authors · 2024-11-28
> **pt 1**
>
> Thanks a lot for your review! It is really thorough and has a lot of actionable feedback. We have incorporated your suggested references, as well as your comments about game dynamics, into the revised paper. To address a couple of your concerns:
>
> > This is a bit underwhelming and entirely hinges on the fact that your deal or no deal game is difficult to play using prompts alone. I am not convinced of this. I am especially not convinced that a better prompt could solve the cooperative game with GPT-3.5.
>
> We’d like to clarify that DoND is actually somewhat hard for LLMs to play. To make this point, we ran additional experiments with our base models on 10 different cooperative and semi-competitive prompts (20 prompts total), devised by the authors. For each prompt, we ran self-play for 100 games; across all prompts, the average self-play score was below 1.2. We added a brief description of this experiment to Appendix B.
>
> We think the reasons for this are twofold: (1) language models like GPT-3.5 struggle with agent tasks, especially without few-shot examples. This is reflected in other agent benchmarks, e.g., [1]; and (2) the fact that proposals are private in DoND means that it is difficult for models to accidentally “stumble into” good behavior. Models need to exhibit good understanding of the task, basic dialogue and math reasoning skills, and the ability to submit formal proposals correctly, to have a reasonable chance at obtaining a nonzero score.
>
> > There should be no communication in a fully competitive game
>
> We generally agree with this point, and we’ve updated our paper in a couple places to reflect it. You are correct that between rational agents, a 2p0s game does not involve any communication (this is partially why the zero-sum results are currently shoehorned into the appendix).
>
> However, the Crawford & Sobel (1982) result does not mean that it is impossible to train models which could in principle use communication to achieve higher scores in competition with humans. Consider, for example, a heads up (i.e., two player) poker game: it’s common for players to speak, and especially if one player is a pro and one player is an amateur, some information can legitimately be gained from this conversation, even though it only tends to benefit the stronger player.
>
> Given that (1) we’re not training models to convergence, (2) these models have a language model prior, and (3) we’re ultimately interested in the performance of models in collaboration or competition with humans, we think it makes sense to keep the results for the zero-sum condition, provided with a disclaimer that communication isn’t useful between fully rational agents.
>
> We also want to provide a couple of clarifications:
>
> > The competitiveness of DoND depends on the specific items and preferences each player has over the items. A game with λ=1 can be highly competitive if both players have similar values for the same items, highly cooperative if they value different items, or a mix of the two.
>
> Perhaps there is a slight mistake here? With λ = 1, both players’ rewards are identical (R_1 = X + 1*Y = 1*X + Y = R_2), therefore the game is fully cooperative. We do agree that this statement holds true when λ = 0, however, and added average Pareto-optimal scores for the semi-competitive settings as you requested
>
> > As well, the that your game is "cooperative, competitive, or anything in between" in your abstract is not necessarily true.
>
> We believe this is shown in the “game objectives” paragraph of Section 2. The game is fully cooperative when λ = 1 (because the rewards are the same), fully competitive when λ = -1 (because the rewards sum to zero), and can be smoothly interpolated in between these settings by changing the value of λ. Please let us know if you have further followup questions here!
>
> [1] AgentBench: Evaluating LLMs as Agents. https://arxiv.org/abs/2308.03688

---

> ### Author Response · Authors · 2024-11-28
> **pt 2**
>
> To answer some questions:
>
> > why do errors occur in 22% of cooperative games but only 1% of semi-competitive? (section 6.1)
>
> We don’t have a clear intuition for why there’s a discrepancy here, but almost all of these errors are of the same type. In the cooperative setting, we found that models often output both a message and a proposal at the same time; when this happened, the environment would throw an error message, giving the model a chance to correct itself. However, in 22% of cases, the baseline model failed to correct itself, resulting in the game aborting. (Note: the reported %s are abort rates; the error rates are higher for both conditions, cf. Figure 10.) Interestingly, we found that this error persisted even across a range of prompts for the cooperative setting.
>
> > how can you be sure that human-AI performance is not bolstered by humans compensating for your AI playing badly?
>
> This is totally possible, and in fact probably true: we suspect that this explains the high initial score for the baseline cooperative model w/ humans (cf. last paragraph of Section 5.1). However, since the performance of the human+LM team still improves over the course of self-play training, we don’t believe that this contradicts our main claims.
>
> > Why do you include lying about an item preference value in your errors for 6.2? Lewis et al argued that this lying could be advantageous / effective deception.
>
> We include this because (1) it is fundamentally difficult to distinguish whether a model is lying strategically or simply hallucinating, and (2) we found during manual annotation that lying about item preference values almost always leads to failed proposals (i.e., scores of zero). In other words, we do not observe cases of GPT-3.5 (or its finetuned variants) using lying productively.

---

> ### Comment · Reviewer_eGtk · 2024-12-01
> **Counterexperiment: GPT-4o mini plays DoND**
>
> A fundamental assumption of this paper is that the DoND game cannot be played by modern language models. Through a small amount of effort, I think I have put this assumption in doubt and I am very unsure about the soundness of this paper. The authors really need to address this point.
>
> Here is an anonymized jupyter notebook where I create a DoND setup and successfully get cooperative DoND negotiation using GPT-4o mini ~https://nbviewer.org/urls/file.io/tAjuZj7zVzxo~ EDIT: see link below
>
> The differences from your setup are
> 1. I used a slightly modified prompt to indicate the game is cooperative and added a tiny amount of chain of thought to improve getting the final proposal in the right format. I also explain what "score" is better than the authors' original prompt
> 2.  I use GPT-4o mini instead of GPT-3.5 turbo
>
> Overall, GPT-4o mini achieves ~60% optimality with minimal prompt tuning in around 2 hours of total work. This is not better than the authors' final self-play result but the story is no longer sound.
>
> These results imply that either the authors assumed a bad prompt (I did a very small amount of prompt tuning) or GPT 4o-mini solves all issues from GPT-3.5. GPT4o-mini was released July 18, well before the submission deadline, and I'm not really sure what is the point of this research direction if a newer models + better prompt does so well. You can make the point that self-play training is easier than prompt engineering but you don't make this point, you claimed it was essentially impossible to play this game with prompting. There is still a point to be made and I still like the research direction but I think the point being made needs to be clear and sound.
>
> **Remaining Concerns**
>
> - The story is still a jumble.
>
> - Thank you for the average scores. Can the authors also add the maximum scores for the semi-competitive setting. A line showing the average and top baselines on Figure 1 would be useful.
>
> - The references to Cao et al and Noukhovitch et al feel minimal if the point you're trying to make is about successful negotiation in a competitive game
>
> - The game is specifically not "*anything* in between [cooperative and competitive]".  The reason for explaining the sampling of values is to specify that it is specifically some level of competitiveness and that you do not provide a general method to get any level of competitiveness (as Noukhovitch et al do).
>
> Overall, I do not feel my concerns are addressed and perhaps I have bigger concerns now that I see GPT 4o-mini can play the cooperative DoND game.

---

> > ### Author Response · Authors · 2024-12-01
> >
> > Thanks for the effort you put in here! Unfortunately, we think these experiments miss the point in a few critical ways:
> >
> > 1. GPT-4o-mini is a substantially stronger model than GPT-3.5 [1]. Nowhere in the paper do we claim that no model would be able to solve DoND without self-play; we simply show that self-play can be used to increase the performance of an existing model.
> > 2. You are correct that GPT-4o-mini was released on July 18th, but public finetuning access (which is required for experiments on self-play training) was not released until later. Our experiments were run before this date.
> > 3. Ultimately, our goal isn’t to build models that can perform well on DoND, but to analyze the effectiveness of self-play as a method for posttraining language models. If we had access to GPT-4o-mini and run experiments on that, then we would have designed a harder task where the base model didn’t perform so well
> >
> > > Can the authors also add the maximum scores for the semi-competitive setting. A line showing the average and top baselines on Figure 1 would be useful.
> >
> > We already included maximum scores for a single game instance in Section 2 ("game setup"), and maximum average scores for a self-play agent in Footnote 1. Can you let us know if you are looking for something else? Also, for the figure, did you mean to say Figure 2? We didn't include horizontal baselines because the figure includes both semi-competitive and cooperative results, which are on different scales, but we could split these into two separate side-by-side figures and include the baselines in the subfigure if you think that would be helpful
> >
> > P.S. The link the Jupyter notebook is broken, and returns a 404 error for us. Could you share a public link?
> >
> > [1] https://openai.com/index/gpt-4o-mini-advancing-cost-efficient-intelligence/

---

> > > ### Comment · Reviewer_eGtk · 2024-12-02
> > >
> > > The original jupyter link expired. Here is an [updated link](https://nbviewer.org/urls/filetransfer.io/data-package/rMPw6rcC/download) that should last for 20 days or 50 views (whichever comes first)
> > >
> > > **overall message**
> > >
> > > It seemed reasonably important that DoND wasn't solvable by LLMs in your previous comment. It feels like backtracking to say that this is just about GPT-3.5 and GPT-4o mini results aren't important. So what is the overall message of your paper? Because it feels quite narrow:
> > >
> > > self-play can be used to learn the rules of a game for an LLM
> > > - but only if the model isn't already too good (GPT 4o-mini) then prompting is sufficient
> > > - and only if the model isn't too bad (LLaMA 70B) then self-play doesn't work
> > >
> > > This feels somewhat obvious and not deep enough to be novel, especially given how similar your "self-play" is to existing methods like ReST-EM.
> > >
> > > **issues with conclusion**
> > >
> > > To understand the point of this paper, I re-read your conclusion and I don't find enough novelty.
> > >
> > > > Our experiments showed that language model self-play can lead to significant performance improvements in both semi-competitive and cooperative games. This finding contradicts existing wisdom that self-play is ineffective in collaborative domains (Strouse et al., 2021), or that models need to be trained on task-specific human data to avoid divergence from human-interpretable language (Lewis et al., 2017; FAIR, 2022).
> > >
> > > The "existing wisdom" you cite isn't about LLMs
> > > - Strouse et al, 2021 show that self-play *from scratch* leads to overfitting to self-play opponents for multi-agent RL games.
> > > - Mitigating alignment tax / divergence from human language is a well known area that you do not tackle at all. Pre-LLM approaches did suffer from divergence [(Lazaridou et al, 2020)](https://arxiv.org/abs/2005.07064) but its not the same post-LLM!
> > >
> > > In the LLM era, your results are easily in line with related work
> > > - RLAIF and RLHF are examples of models finetuning with each other and generalizing to humans
> > > - Language divergence is still happening, even in your case! In RLHF literature this is known as "alignment tax" or "reward model overoptimization"
> > > -  Despite alignment tax, it is well known that models can RL finetune to a task (e.g. coding, math) while maintaining human-interpretable language
> > >
> > > **summary**
> > >
> > > Overall, I don't think this paper should be accepted.
> > > - If the goal is to analyze self-play as a method, then it makes sense to compare it to other methods for post-training. But there are not really any comparisons
> > > - DoND doesn't feel like a difficult challenge if it is solvable with GPT-4o mini and prompting.
> > > - I just don't see a novel, coherent story to the results presented, and the overall results do not contradict existing wisdom in my opinion.
> > >
> > > I believe there is a coherent and interesting paper to be made but you haven't made the major changes necessary to make this a compelling story or reasonable conclusion

---

### Comment · Area_Chair_dV42 · 2024-11-29

Dear Reviewers,

This is a friendly reminder that the last day that reviewers can post a message to the authors is Dec. 2nd (anywhere on Earth). If you have not already, please take a close look at all reviews and author responses, and comment on whether your original rating stands.

Thanks,

AC

---

### Meta-Review · Area_Chair_dV42 · 2024-12-20

**Metareview:**

This paper explores the application of self-play training to language models in non-zero-sum settings through experiments in the Deal or No Deal negotiation game.

The reviewers agree that the proposed self-play training on large language models is a novel approach. However, the current version of this paper has the following weaknesses.
- The study lacks evaluations across additional environments or tasks to generalize its findings. Thus, claims of large performance improvements might be overstated given the restricted task, model, and training setup.
- Does not include reinforcement learning (RL) baselines for benchmarking.
- Limited insights into why self-play works in some cases and fails in others.

**Additional Comments On Reviewer Discussion:**

The reviewers reach an agreement during the rebuttal.

---

### Decision · Program_Chairs · 2025-01-22

Reject